# Laherradurin Inhibits Tumor Growth in an Azoxymethane/Dextran Sulfate Sodium Colorectal Cancer Model In Vivo

**DOI:** 10.3390/cancers16030573

**Published:** 2024-01-29

**Authors:** Michael Joshue Rendón-Barrón, Eduardo Pérez-Arteaga, Izamary Delgado-Waldo, Jossimar Coronel-Hernández, Carlos Pérez-Plasencia, Frida Rodríguez-Izquierdo, Rosa Linares, Alma Rosa González-Esquinca, Isela Álvarez-González, Eduardo Madrigal-Bujaidar, Nadia Judith Jacobo-Herrera

**Affiliations:** 1Unidad de Bioquímica, Instituto Nacional de Ciencias Médicas y Nutrición Salvador Zubiran, Av. Vasco de Quiroga 15, Col. Belisario Domínguez Sección XVI, Tlalpan, Ciudad de México 14080, Mexico; mrendonb2101tmp@alumnoguinda.mx (M.J.R.-B.); eduardo.arteaga@comunidad.unam.mx (E.P.-A.); mybtondgomita@comunidad.unam.mx (I.D.-W.); 2Unidad Profesional Adolfo López Mateos, Laboratorio de Genética, Instituto Politécnico Nacional, Escuela Nacional de Ciencias Biológicas, Zacatenco, Av. Wilfrido Massieu Esq Cda. Miguel Stampa S/N, Alcaldía Gustavo A. Madero, Ciudad de México 07738, Mexico; ralvarezg@ipn.mx (I.Á.-G.); emadrigalb@ipn.mx (E.M.-B.); 3Unidad de Investigación Biomédica en Cancer, Instituto Nacional Nacional de Cancerología, Av. San Fernando 22, Belisario Domínguez Secc 16, Tlalpan, Ciudad de México 14080, Mexico; jossithunders@gmail.com (J.C.-H.); carlos.pplas@campus.iztacala.unam.mx (C.P.-P.); fridaizquierdo19@gmail.com (F.R.-I.); 4Unidad de Investigación en Biomedicina, Laboratorio de Genómica, Facultad de Estudios Superiores Iztacala, Av. De los Barrios 1, Los Reyes Ixtacala, Tlalnepantla 54090, Mexico; 5Unidad de Investigación en Biología de la Reproducción, Laboratorio de Endocrinología, Facultad de Estudios Superiores Zaragoza, Batalla 5 de Mayo S/N, Ejército de Oriente Zona Peñon, Iztapalapa, Ciudad de México 09230, Mexico; rosa.linares@zaragoza.unam.mx; 6Laboratorio de Fisiología y Química Vegetal, Instituto de Ciencias Biológicas, Universidad de Ciencias y Artes de Chiapas, Libramiento Norte Poniente 1150, Lajas Maciel, Tuxtla Gutiérrez 29039, Mexico; aesquinca@unicach.mx

**Keywords:** acetogenins, antitumoral activity, apoptosis, cell migration, colon cancer, in vivo model, laherradurin

## Abstract

**Simple Summary:**

The complexity of the tumor cell, its ability to evade death, and its capacity to proliferate and conquer new tissues and organs make its treatment a challenge. Major challenges posed by therapeutic agents include their toxicity and consequences for patients after chemotherapy. Natural products provide a fundamental basis for the development of chemoagents. Our research aimed to test in vivo the antitumor activity of laherradurin in CRC and to measure its toxic effect. Our findings show that laherradurin inhibits growth and decreases the size of tumors in the colon compared with cisplatin. In addition, blood tests show no hepatic or renal damage or damage at the immunological level. Likewise, the disease activity index shows that those animals treated with laherradurin had diminished signs of disease. In our work, we provide new and conclusive evidence on the antitumor effect of laherradurin and measure its toxic activity under different criteria.

**Abstract:**

Colorectal cancer (CRC) is the third most common neoplasia in the world. Its mortality rate is high due to the lack of specific and effective treatments, metastasis, and resistance to chemotherapy, among other factors. The natural products in cancer are a primary source of bioactive molecules. In this research, we evaluated the antitumor activity of an acetogenin (ACG), laherradurin (LH), isolated from the Mexican medicinal plant *Annona macroprophyllata* Donn.Sm. in a CRC murine model. The CRC was induced by azoxymethane–dextran sulfate sodium (AOM/DSS) in Balb/c mice and treated for 21 days with LH or cisplatin. This study shows for the first time the antitumor activity of LH in an AOM/DSS CRC model. The acetogenin diminished the number and size of tumors compared with cisplatin; the histologic studies revealed a recovery of the colon tissue, and the blood toxicity data pointed to less damage in animals treated with LH. The TUNEL assay indicated cell death by apoptosis, and the in vitro studies exhibited that LH inhibited cell migration in HCT116 cells. Our study provides strong evidence of a possible anticancer agent for CRC.

## 1. Introduction

Colorectal cancer is a health problem worldwide, being the third most diagnosed type of cancer, with more than 1.9 million new cases and 900 thousand reported deaths per year since 2020, being the second leading cause of cancer-related death [1,2]. Current treatments consist of surgery and preoperative radiotherapy for patients in early and intermediate stages, while in stage III and metastatic patients, chemotherapy is mainly used [3,4]. The chemotherapy used in CRC includes 5-fluorouracil, capecitabine, irinotecan, oxaliplatin, trifluridine, and tipiracil [5]; however, they have severe side effects in patients, such as neurological damage, cytopenia, and mucositis [6,7,8,9]. Moreover, the high mortality rate is related to drug resistance and late diagnosis [10].

The search for molecules from natural products is not new and throughout history has been a successful approach to developing drugs for various diseases, including cancer [11]. Some specialized pharmacologically active plant metabolites are lignans, glycosides, diterpenes, steroids, fatty acids, flavonoids, and acetogenins (ACGs) [12,13]. The latter is selectively toxic to several types of cancer cells and multidrug-resistant cell lines [14,15], which interest our group. ACGs are compounds found in species of the Annonaceae family, particularly in the *Annona* genus; the fruits of these plants are employed in traditional medicine for various ailments, such as fever, rheumatism, diarrhea, arthritis, diabetes, headaches and insomnia, and cancer treatment [16,17]. The leading studies focus on ACGs from *A. muricata*, reporting their anti-inflammatory, pesticidal, antimicrobial, and mainly antitumor activities [18]. Therefore, it is necessary to study ACGs in other species of the *Annona* genus, such as *A. macroprophyllata* Donn. Sm., which is traditionally employed in Mexico for the treatment of skin cancer and gastric tumors [19,20], in addition to reporting the presence of LH (Figure 1). LH is a cytotoxic acetogenin proven effective against cervical and colon tumor lines [21]; even so, studies in vivo of its antineoplastic potential are limited. Therefore, in this work, we evaluated the anticancer effect of LH in vivo in a colon cancer model by chemical induction with AOM and DSS.

## 2. Results

### 2.1. Acute Toxicity (LD_50_)

To formulate safe doses to be used in animals, we performed an acute toxicity test based on the OECD criteria (Section 4). Animals administered 55 and 175 mg/Kg LH presented high toxicity since none of the experimental groups survived. As for the dose of 17.5 mg/Kg, only one death was recorded, while the mice treated with 5.5 mg/Kg survived the LH. Based on the above and the formula described, the LD_50_ was 11.5 mg/Kg. This dose was employed to determine the doses used in the AOM/DSS model (Section 2.2.).

### 2.2. AOM/DSS Model

#### 2.2.1. Weight and Disease Activity Index (DAI)

Toxicity in new anticancer drugs is vital and key to the possibility of using them in clinical settings. Our research evaluated the toxicity of LH from two different and complementary perspectives: the macroscopic level, measuring weight and signs of illness (DAI) during the 16 weeks of the in vivo trial, and blood tests (Section 2.2.2.). Figure 2A depicts the average weight of each experimental group. In general, mice gained between 1.5 and 2.5 g compared with the negative control group (NC). However, from week 12 (start of LH treatment) until the end of the trial, weight recovery was observed in the groups treated with LH. Depending on the doses, the percentage of weight recovery was low (17.6%), medium (36.37%), high (46.10%), and high twice weekly (51.74%). In the cisplatin group, the weight recovery was 38.11%. Figure 2B shows the DAI results. The groups treated with AOM/DSS at week 11 presented values of 1.5 and 1.8 compared with the NC group. By week 16, the positive control group (PC) presented the highest value, with more severe disease symptoms (bloody diarrhea and anal prolapse). Groups treated with LH displayed fewer disease symptoms, given in percentages (DAI values). As the dose and frequency of administration were increased, the animals treated with LH saw improved effects of the drugs on cancer symptoms, as follows: low dose (30.8%), medium dose (56.4%), high dose (92.4%), and the twice-a-week high dose (90.6%). As for cisplatin-treated mice, the DAI values decreased by 53.1%. These data suggest lower toxicity compared with cisplatin.

#### 2.2.2. Blood Analysis

Continuing with the toxicity analyses, we conducted blood tests when treatments finished as an end-point experiment. Table 1 shows the results of the biochemical blood analysis of the AOM/DSS model. The CP group presented a decrease in the levels of erythrocytes, hemoglobin, leukocytes, and ALT, as well as in the percentages of hematocrit, lymphocytes, and neutrophils. Moreover, the urea and glucose values increased compared with the negative control, presenting significant differences (*p* ˂ 0.05). Regarding the groups treated with LH, remarkably, in those groups treated with the high dose (3 mg/Kg) either one or two times per week, the values of erythrocytes, hemoglobin, hematocrit, ALT, urea, and glucose were similar to those in the CN group. In the group treated with cisplatin, significant differences (*p* ˂ 0.05) were observed in the levels of erythrocytes, hemoglobin, and ALT and the percentage of hematocrit. Our data support the benefits of LH over cisplatin in terms of the effects of toxicity on our AOM/DSS model.

#### 2.2.3. Macroscopic Analysis of the Colon of Balb/c Mice

To complete the macroscopic examination of the colon, we considered the number and size of tumors, as well as the length of the colon. Figure 3 shows pictures of the colons of each of the experimental groups. Figure 3A depicts a healthy colon, whereas Figure 3B illustrates the AOM/DSS group (presence of tumors). The groups treated with LH (Figure 3C–F) exhibited fewer and smaller tumors. The cisplatin group (Figure 3G) displayed similarities to the group treated with the medium dose of LH (Figure 3D).

Figure 4A displays the results for the number of tumors, where the PC presented an average of 33 tumors. The groups treated with LH showed decreases in the tumor index of 13.6%, 45%, 66.6%, and 69.7% for the low, medium, and high doses once and twice a week, respectively. The group administered cisplatin showed a decrease of 40.16% compared with the PC. Figure 4B presents the results for tumor size, where the same behavior can be observed, since the highest percentages of reduction were observed in the groups administered the high dose of LH (3 mg/Kg) once and twice a week, showing decreases of 72.5% and 78.34%, respectively. Regarding colon size (Figure 4C), the PC group presented a 35% decrease in length with respect to the NC, while the groups treated with the medium and high doses of LH one and two times per week avoided shortening, as there were decreases of 28%, 14.5%, and 14%, respectively. The group treated with cisplatin showed a decrease of 27%, showing similarity to the group administered the 1.5 mg/Kg dose of LH. The size of the colon and the significant reduction in tumors suggest the antitumor effect of LH in vivo.

#### 2.2.4. Histology Image Analysis

Figure 5A shows a longitudinal section of the colon of a healthy animal. It is characterized by a muscular layer with circular and longitudinal fibers, a mucosal layer composed of glands, which form the crypts and connective tissue between them, and the lamina propria, which gives way to the submucosal layer. At 400× magnification, the surface epithelium and the crypt epithelium can be appreciated. On the contrary, in the AOM/DSS colon tissue that was not treated (Figure 5B), tumor cells are in the form of tubular adenomas with a high degree of dysplasia, loss of a healthy crypt organization, and the absence of a simple columnar epithelium observed. In animals treated with LH at 0.5 mg/Kg (Figure 5C), the image shows the presence of tumor cells and the loss of the mucosa layer, the surface epithelial tissue, and the crypts. Figure 5D depicts a longitudinal section of the colon treated with LH at 1.5 mg/Kg. In this image, the mucosal layer can be seen to be recovered, together with the presence of glands forming the crypts, although still without epithelial tissue inside them (magnification at 400×). Recovery of the lamina propria, a muscular layer with circular muscle fibers, and surface epithelial tissue can also be seen. In the experimental groups treated with LH at 3 mg/Kg once or twice a week (Figure 5E,F), a rearranging of the epithelial tissue (head of the arrow) of the crypts and the mucosa was observed. The mucosa is composed of glands with connective tissue between them and the lamina propria. At the end of this layer is the submucosa, formed of connective tissue, and the muscular layer with circular and longitudinal muscle fibers. Finally, cisplatin treatment restored the epithelial tissue, crypts, mucosa, and submucosa (Figure 5G).

#### 2.2.5. Apoptosis Induction by LH

Apoptosis evasion is one of the hallmarks of cancer. Thus, searching for cell death induction is a goal of cancer treatment. For this research, we evaluated the presence of apoptotic bodies in the tumors extracted from the colon after being treated with LH or cisplatin. The LH-induced DNA fragmentation was visualized by confocal microscopy using a TUNEL assay. Figure 6 depicts the colon tissue. In the PC group (saline solution), in the third column, no apoptotic bodies were found. On the other hand, the LH-treated groups (columns 5–8) showed DNA fragmentation in different regions of the tissue. The apoptotic bodies’ presence depends on the doses; the greater the dose, the greater the apoptosis induction. Our results show the ability of LH to induce cell death by apoptosis in the AOM/DSS CRC model.

#### 2.2.6. Inhibition of Migration (Wound Healing Assay)

As known, metastasis is one of the major complications for cancer patients. Thus, we evaluated another probable effect of LH on one of the hallmarks of cancer cells, migration to other tissues or organs. To this end, we carried out a wound healing assay in vitro on the CRC cell line HCT116. Figure 7A shows the panel of photos of the cells, while Figure 7B shows the graph of the migration percentages of each experimental group. After 24 h post-stimulus, the LH-treated group inhibited cell migration by preventing wound closure (14.60%) compared with the untreated group (SFB 10%), where wound closure by cell migration was 41.13%. Such results indicate another possible target for LH due to its capacity to stop cells from migrating to other tissues. However, more experiments should be performed to confirm and strengthen such findings.

## 3. Discussion

Throughout history, people have used nature to alleviate different ailments, with plants and their secondary metabolites playing a central role [22] due to their biological activity and ability to improve human health [23]. Currently, 24% of drugs prescribed are derived from plant species [24], mostly from medicinal plants. Such is the relevance of traditional medicine that the World Health Organization, in the year 2022 in India, established the WHO World Center for Traditional Medicines with the objective of linking modern science with the traditional health system and fostering its evolution in favor of health, preserving natural resources, and respect for local heritage, among other factors (WHO, 2023) [25]. In addition, the First WHO World Summit on Traditional Medicinal Products was held in August of this year. In this scenario, the current importance of the use of medicinal plants in different societies is supported, either as a first line of treatment or as a complement to therapies, which is why evaluating the properties of specialized metabolites, extracts, and herbal combinations is of relevance to validate their use, evaluate their toxicity, determine their mechanisms of action, propose the repositioning of their use in other diseases, and formulate new drugs. Several studied specialized plant metabolites have antineoplastic activity. Among them, acetogenins are of particular importance, as they are selectively toxic to several types of cancer cells and to cell lines resistant to multiple drugs [15,26,27,28,29,30,31,32].

Regarding the antitumor activity of ACGs in vivo, to our knowledge, it has only been reported in xenograft models as an inhibitor of tumor growth [15,21,33,34,35]. In this work, the antineoplastic activity of LH isolated from the seeds of *A. macroprophyllata* was evaluated for the first time in an AOM/DSS-induced colon carcinogenesis model. Toxicity in cancer patients is a priority concern, as oncological drugs produce severe side effects that weaken the patient and can trigger other complications that delay recovery [36,37,38]. In this study, we determined the LD_50_ of LH to define its toxicity, obtaining a concentration of 11.5 mg/Kg, which is classified as moderate toxicity according to the OECD [39]. Based on this concentration, the experimental doses for the carcinogenesis assay were established. Toxicity studies in complete organisms are necessary to assess the parameters of damage at the systemic level of new possible anticancer agents. Little is known about the toxicity of ACGs, although the extract of A. *macroprophyllata* is cytotoxic at very low doses [40], and in species such as *A. muricata* and *A. squamosa*, they exhibit neurotoxicity [16,41,42]. Another indicator of toxicity is weight monitoring in experimental animals. Malnutrition is a common problem in CRC patients due to intestinal obstruction and cachexia [43] and has been associated with overexpression of cytokines such as interleukin-6 and tumor necrosis factor-alpha (TNF-α), as well as hormones that induce lack of appetite, and it has direct catabolic effects on skeletal muscle [44,45,46]. In a recent study, weight loss of more than 20% was observed in a colon cancer model induced with AOM/DSS [47], which is consistent with our results since the PC group presented a 31% decrease compared with the negative control. In addition to weight loss in animals, this model has been associated with other symptoms, such as severe colitis, diarrhea with bleeding, and the development of tumors in the colon [48]. Therefore, the DAI score allowed us to evaluate the progression and severity of the disease during a 16-week trial, where our results show that the AOM/DSS-treated groups presented symptoms of diarrhea and bleeding compared with the NC group. DSS in mouse models causes chronic inflammation and ulceration of the colon due to macrophage dysfunction, changes in intestinal microflora, and a toxic effect on the intestinal epithelium [49,50,51,52].

Concerning blood tests, we observed that the groups treated with AOM/DSS presented statistically significant changes in the levels of hemoglobin, erythrocytes, leukocytes, lymphocytes, hematocrit, ALT, urea, and glucose compared with the group of healthy mice. However, the groups administered LH showed less toxicity than the groups treated with cisplatin. Our results for weight loss and the decrease in hemoglobin levels could be related to the development of anemia in the animals. Bone marrow involvement, tumor-associated blood loss, and iron deficiency due to overexpression of cytokines such as IL-6 have been shown to cause the occurrence of chronic anemia in cancer patients [53,54,55]. On the other hand, blood data could act as biomarkers in evaluating and predicting gastric cancer [54]. The antineoplastic effect of LH was macroscopically assessed by considering the number of tumors, their size, and the colon length of each of our experimental groups. The results in Figure 5A indicate that the groups administered 3 mg/Kg LH once or twice weekly demonstrated significant decreases in the tumor index of 66.6% and 69.7%, respectively, compared with the drug cisplatin with a 40.16% decrease. Similar results were observed for the reduction in tumor size, with the dose of 3.0 mg/Kg being more effective than cisplatin. Other works report that mice administered AOM/DSS show colon shortening, changes in morphology, and tissue stiffness [47,56,57], agreeing with our results since the groups with induced CRC showed a decrease in colon length compared with the group of healthy mice. The histopathology results obtained by hematoxylin–eosin staining suggest that those mice administered LH displayed restoration of the damaged mucosa layer, recovery of the lamina propria, and the absence of adenoma and intramucosal carcinoma. In addition, hyperchromatic epithelial regeneration was observed in the crypts. Such results are similar to the effect caused by cisplatin. It is still not known how ACGs inhibit morphological changes in the tumor cell; however, the upregulation of pro-inflammatory cytokines can control tumor formation in the colon [57,58].

According to the abovementioned results, we observed that the dose of 3 mg/kg presented a greater antineoplastic effect in comparison with cisplatin, with significant differences (*p* ˂ 0.05). In addition, our results indicate a decrease in toxicity at the macroscopic level in the animals and in the blood tests, which were evaluated with liver and renal damage enzymes. The evaluation of LH toxicity is needed because drugs used as antineoplastic agents have adverse effects. Such is the case of cisplatin, which causes mainly nephrotoxic damage, with over 30% of patients presenting acute and chronic renal lesions [59]. Moreover, cisplatin could induce neurotoxicity, hepatotoxicity, ototoxicity, and testicular damage [60,61,62,63].

The neurotoxicity of acetogenins is reported in several works [64,65,66,67]. Some works demonstrated neural cell death and a change in the intracellular distribution of tau in primary cultures produced by annonacin. When the protein tau accumulates in cell bodies, it induces neurodegenerative problems called tauopathies [66]. These studies point out the neurotoxicity of the acetogenin annonacin and the neuronal damage caused in a tropical community where the consumption of this plant is a common practice [64,65]. However, it is necessary to investigate the form of preparation, the frequency of ingestion, and whether the damage caused is the same as that observed in other pathologies, such as cancer. Therefore, the antecedents of toxicity in acetogenins deter us from the possibility of inducing brain damage, and further studies are necessary.

Acetogenin cytotoxicity is explained by the inhibition of the mitochondrial NADH-ubiquinone reductase complex (complex I) of the electron transport chain involved in ATP synthesis. This complex could be a potential target in cancer treatment, given that cancer cells have a higher demand for ATP than normal cells [18,68,69]. Furthermore, several authors have found that ACGs induce cell death by apoptosis [70,71,72]. For instance, annonacin and squamocin induce apoptosis in several types of cancer throughout the upregulated expression of caspase 3, 8, and 9, and in turn, the downregulation of Bcl-2 [33,73,74,75], which is in agreement with De Pedro et al. (2013) [76], who observed an increase in caspase 9 activations in LH-treated cell lines. In this work, a TUNEL assay was carried out to determine whether LH induced apoptosis. The LH-treated groups showed gradual DNA fragmentation, with the high double dose inducing the most apoptosis. This result confirms the capacity of LH to induce cell death in vivo by apoptosis as a mechanism of action.

On the other hand, we report for the first time in this work the capacity of LH as an inhibitor of HCT116 tumor cell migration in vitro. This result suggests that it is possible to use this acetogenin to prevent tumor cells from colonizing tissues and organs to generate metastasis. In CRC, the most frequent organ of metastasis is the liver; 50% of patients develop it in the course of the disease [77], and those patients with gastrointestinal tumors (including the colon) have a lower probability of survival if they develop metastasis to other organs [78]. Regarding the genus *Annona*, the extract of the species *A. muricata* was the only one reported to have antimetastatic properties in an orthotopic model of pancreatic tumor cells [15]. The process of metastasis is complex, involving not only genetic aberrations, vascular growth, metabolic reprogramming, and resistance to therapy but also factors involving the pro-inflammatory microenvironment in CRC [77]. Our migration results are not definitive but suggest the possibility of using LH as an antimetastatic drug.

## 4. Materials and Methods

### 4.1. Chemicals

Dextran sulfate sodium (DSS PM. 36,000–50,000 M W) MP Biomedicals (Irvine, CA, USA), antibiotic Capricorn Scientific (Ebsdorfergrund, Germany), silica gel column 60 HF Merck^®^ (Darmstadt, Germany) were used in this study. The substances purchased from Sigma-Aldrich (St. Louis, MO, USA) included Azoxymethane CAS 25843-45-2 (AOM), phosphate-buffered saline P3813-10PAK (PBS), and trypsin. Fetal bovine serum (FBS) and RPMI 1640 medium were obtained from Gibco^TM^ (Billings, MT, USA), and organic solvents (hexane, ethyl acetate, and methanol) were obtained from Baker^®^, Phillipsburg, NJ, USA.

### 4.2. Isolation and Identification of LH 

LH was isolated from the seeds of *A. macroprophyllata* Donn.Sm. and compared with the original LH isolated from the seeds of the same species [21]. The plant was collected in Chiapas, Mexico, and identified with number 352 in the Eizi Matuda Herbarium of the Universidad de Ciencias y Artes de Chiapas, Mexico. Once dried, continuous extractions of the dried seeds were obtained using Soxhlet equipment with hexane (8 h/3 times) and methanol (8 h/3 times) to obtain the hexane and methanolic extracts. The latter was fractionated on a silica gel column (14 × 6 cm, extract/SiO_2_ 1:15, flow rate 20 mL/min). Organic solvents were used with an increasing polarity gradient of Hex, as follows: AcOEt, AcOEt, AcOEt, MeOH, and MeOH. The identification and purity of the LH were determined by high-performance liquid chromatography (HPLC), thin-layer chromatography (TLC), and melting point identification. The chemical characterization of LH was carried out by HPLC (HPLC 200, Perkin Elmer, Norwalk, CT, USA) at 220 nm, with a retention time of 4.2 min, a mobile phase methanol/acetonitrile/water ratio of 10/80/10, and a silica column, Spheri-5 RP-18 (100 × 4.6 mm, 5 µm), at 30 °C, and nuclear magnetic resonance analysis. LH had a melting point of 66–67 °C.

### 4.3. Acute Toxicity Determination of the Mean Lethal Dose (LD_50_)

The LD_50_ was obtained based on OECD criterion 423 in order to establish the experimental doses for subsequent in vivo tests. Balb/c mice were administered doses ordered with a factor of 3.2 (1.75, 5.5, 17.5, 55, 175, 550, and 2000 mg/Kg). The trial was initiated with the 175 mg/Kg dose and terminated when all three animals survived at each dose used. LH was administered intraperitoneally (i.p.). Three organisms were used for each experimental dose (OECD, 2002) [39].

The LD_50_ was calculated according to the criteria of Chinedu et al. (2013) [79] following the formula: LD_50_ = [M_0_ + M_1_]/2;M_0_ = highest dose without mortality;M_1_ = lowest dose with at least one deceased.

### 4.4. In Vivo Model of Colon Cancer

#### 4.4.1. Animals

Forty-two male Balb/c mice, 6 to 8 weeks old and with an average weight of 20 to 25 g, were used and kept in the INCMNSZ animal facilities with appropriate conditions (housing in polypropylene cages, light and dark cycles (12/12 h), temperature of 23 ± 2 °C, water and food ad libitum, following international recommendations related to animal care and handling reported in the ARRIVE 2.0 guidelines [80]. The research protocol was approved by the Animal Ethics Committee of the National Nutrition Institute of Mexico (CICUAL No. BQO-1932-19-21-1).

#### 4.4.2. Induction of Colon Cancer by AOM/DSS

The induction of carcinogenesis was modified from the inflammatory model proposed by Modesto et al. in 2022 [52]. Firstly, mice received a single dose of azoxymethane (AOM) at 12. 5 mg/Kg i.p. Subsequently, the mice received three cycles of sodium dextran sulfate (DSS) at 2% dilution in their drinking water. Each cycle lasted seven days, and there were two weeks of rest between each cycle (Figure 8A).

Animals were randomly organized into seven experimental groups of six individuals each. After induction of carcinogenesis by AOM/DSS, LH treatment started with weekly administration for five weeks (Figure 8B). Group 1, positive control, no treatment; Group 2, LH at 0.5 mg/Kg; Group 3, LH at 1.5 mg/Kg; Group 4, LH at 3.0 mg/Kg; Group 5, LH at 3.0 mg/Kg twice a week; Group 6, cisplatin (reference drug) at 2 mg/Kg; and Group 7, healthy animals for colon reference. The experimental doses of LH were established according to its LD_50_. During the whole experiment, the weight of the animals was recorded weekly and graphed on weeks 11 and 16. DAI data followed the scale values 0: solid feces; 1: soft feces (diarrhea); 2: normal feces with visible blood; and 3: bloody diarrhea [81].

#### 4.4.3. Blood Test and Colon Macroscopic Analysis

One week after the last LH administration, blood collection by cardiac puncture was performed to carry out blood biometry and biochemistry at the Divet Laboratory, Mexico. Afterward, the mice were sacrificed to obtain the colon. The number and size of tumors and the length of the colon were recorded.

### 4.5. Histology and Immunohistochemical Analysis

#### 4.5.1. Hematoxylin–Eosin Staining

To find histological alterations, the colon was examined using 5 mm thick sections with serial sections up to 200 mm and stained with hematoxylin/eosin. The sections were observed with the help of a Nikon Eclipse E400 microscope (Nikon, Tokyo, Japan), a digital Nikon DS-U2 camera (Nikon, Tokyo, Japan), and the software NIS-Elements version 3.0. Histology classification was performed according to recommended nomenclature for intestinal neoplasms in murine models [82].

#### 4.5.2. Terminal Deoxynucleotide Transferase-Mediated Deoxy-UTP Nick End Labeling (TUNEL) Analysis

A TUNEL assay was executed to identify apoptotic cells in the colon tissues of LH-treated mice. Tissue sections were deparaffinized and rehydrated. DNA fragmentation was determined by catalytic incorporation of fluorescein-12-dUTP(a) into the 3′-OH ends of DNA using recombinant terminal deoxynucleotidyl transferase (rTdT) enzyme, according to the manufacturer’s protocol (Dead End Fluorometric TUNEL System, Promega Part # TB235). Finally, the slides were washed with double-distilled water and cover-slipped using a Vecta Sheeld mounting medium (Vector Laboratories, Newark, CA, USA). Images were taken on a Zeiss LSM 880 confocal microscope.

### 4.6. Wound Healing Assay

To evaluate migration inhibition by LH, a wound closure assay was performed in vitro, according to Grada et al. (2017) [83]. In a 6-well plate, 2 mm adhesive tape was placed, and the HTC116 colon cancer cells were seeded at a density of 1.5 × 10^6^ per well. The plates were incubated for 24 h in an atmosphere of 5% CO_2_, 95% humidity, and 37 °C. Subsequently, the tapes were removed, and the cells were treated with LH at a concentration of 8 µM for 24 h. Treated cells were supplemented with SFB at 2%, and control cells with 10% SFB. The experiment was performed in triplicate. Photos of each well were taken with the Nikon TS100 microscope and analyzed with the help of the software ImageJ version 1.53k.

### 4.7. Statistics

Results are presented as the mean ± ES, weight and DAI, and blood and tumor analyses of each experimental group. One-way analysis of variance (Anova) and Kruskal–Wallis with a Newman–Keuls post hoc comparison test (*p* ˂ 0.05) were performed, with the results given as percentages (hematocrit, lymphocyte, and neutrophil blood tests).

## 5. Conclusions

The high rate of CCR in the world population, the mortality numbers, and the lack of efficient treatment, chemoresistance, and metastasis are issues for public health discussion. Natural products are a source of chemical structures with varying bioactivity, presenting an opportunity for developing new drugs. Our research demonstrates the antitumor capacity of the acetogenin laherradurin in a CCR model in vivo. The LH reduced the number and size of the tumors and showed less toxicity at macroscopic and blood levels compared with the reference drug, cisplatin. Moreover, LH induced cell death by apoptosis in vivo and inhibited cell migration in HCT colon cancer cells. Also, those animals treated with LH manifested less toxicity, as indicated by the weight recovery and reduced disease signs compared with animals treated with cisplatin. These results bring new and worthwhile information regarding the potential of laherradurin as an anticancer drug for CCR. The study of this molecule is not over, as questions arising from our results about its activity, other mechanisms of action, possible therapeutic combinations, additional toxicity assays, and signaling pathways involved are under investigation by our research group.

## Figures and Tables

**Figure 1 cancers-16-00573-f001:**
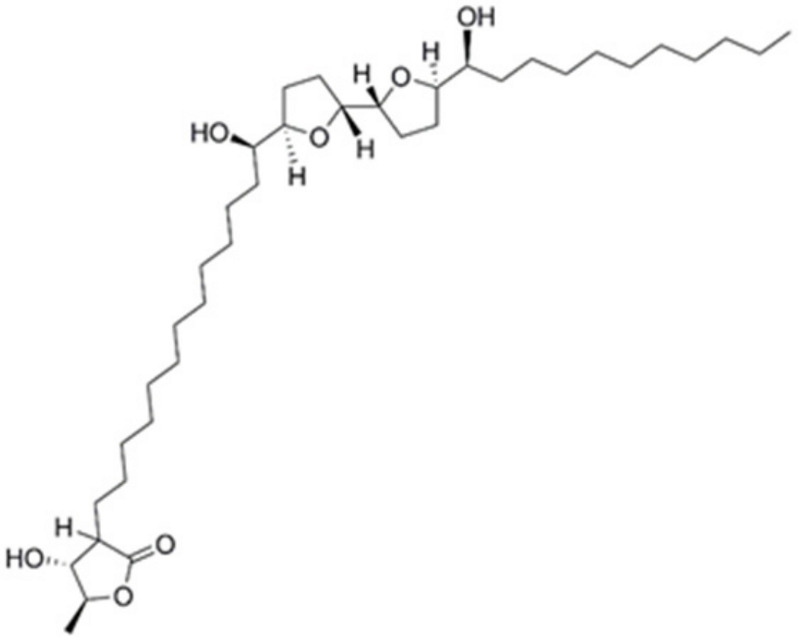
Chemical structure of laherradurin.

**Figure 2 cancers-16-00573-f002:**
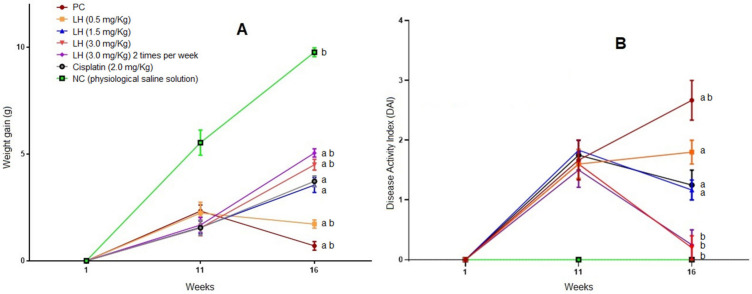
AOM/DSS toxicity data. (**A**) Variations in weight. (**B**) Disease Activity Index data. Data shown are the mean ± SE of six mice per group. a, significant differences against the negative control; b, significant differences against cisplatin. Anova and post hoc Newman–Keuls (*p* ˂ 0.05).

**Figure 3 cancers-16-00573-f003:**
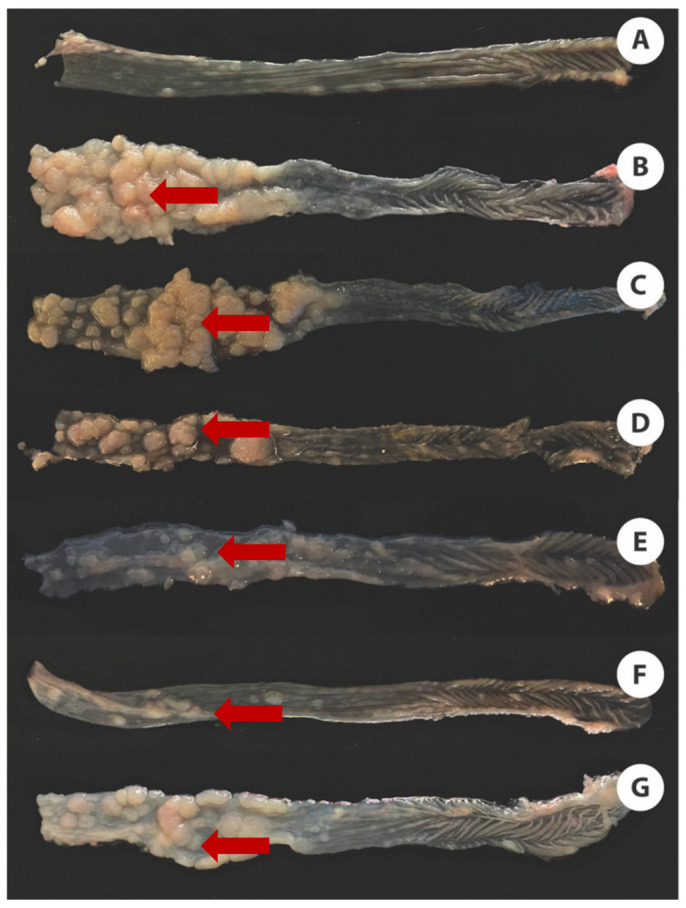
Photos of the colon after AOM/DSS scheme and treatments. (**A**): CN (healthy colon); (**B**): CP (AOM/DSS); (**C**): treatment with LH 0.5 mg/Kg; (**D**): treatment with LH 1.5 mg/Kg; (**E**): treatment with LH 3.0 mg/Kg; (**F**): treatment with LH 3.0 mg/Kg twice a week; (**G**): treatment with cisplatin 2.0 mg/Kg. All groups were administered LH or cisplatin once a week, except those indicated in the Methodology section. Red arrows point to tumors. The pictures are representative of the experimental groups.

**Figure 4 cancers-16-00573-f004:**
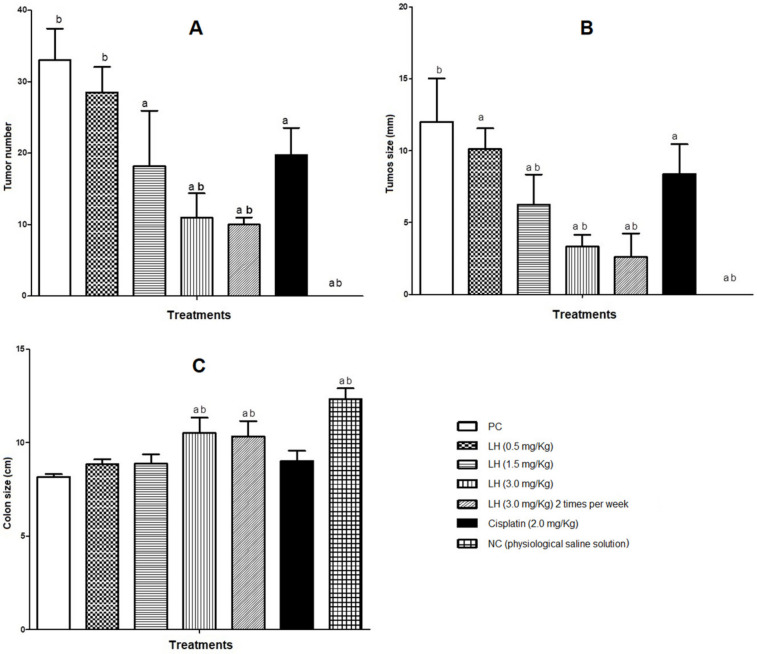
Quantitative analyses of the colon after treatment with LH and cisplatin. (**A**) tumor number, (**B**) tumor size, and (**C**) colon size. Data shown are the mean ± SD of six mice per group. a, significant differences against the positive control; b, significant differences against cisplatin. Anova and post hoc Newman–Keuls (*p* ˂ 0.05).

**Figure 5 cancers-16-00573-f005:**
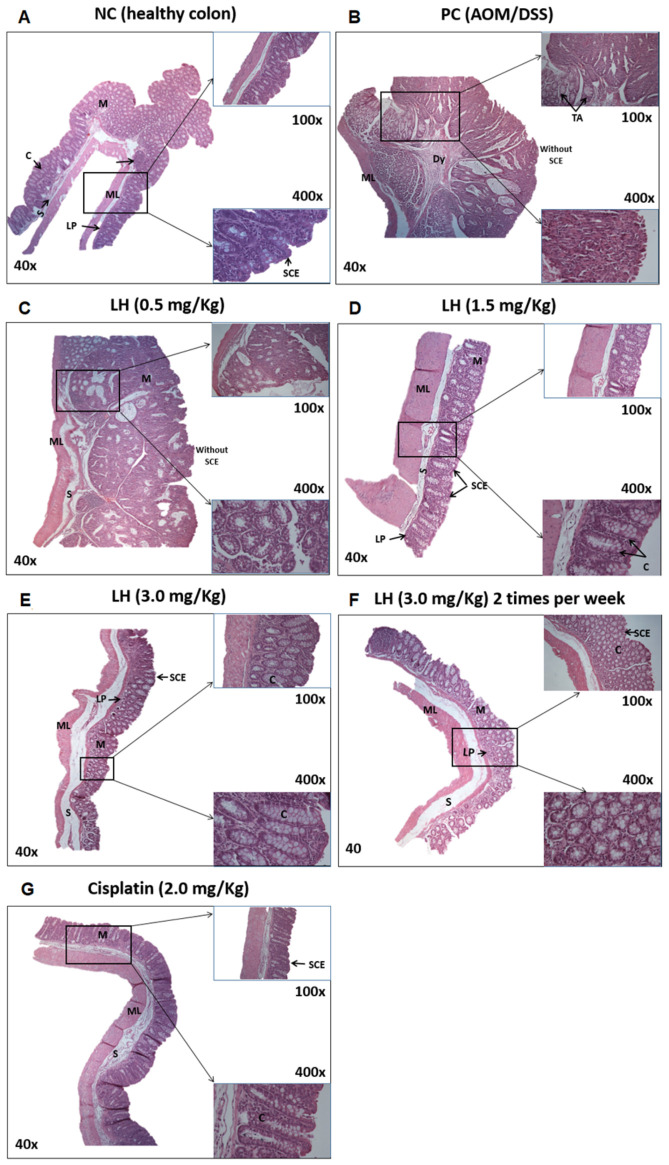
Histology image of the colon (H&E). NC: physiological saline solution, PC: positive control. ML: muscular layer, Dy: dysplasia, TAs: tubular adenomas, LP: lamina propria, SCE: simple columnar epithelium, M: mucosa, S: submucosa, C: crypts. Magnifications 40×, 100×, and 400× were used to amplify the selected section of the tissue cut for examination. Images were taken with a Nikon Eclipse E400 microscope.

**Figure 6 cancers-16-00573-f006:**
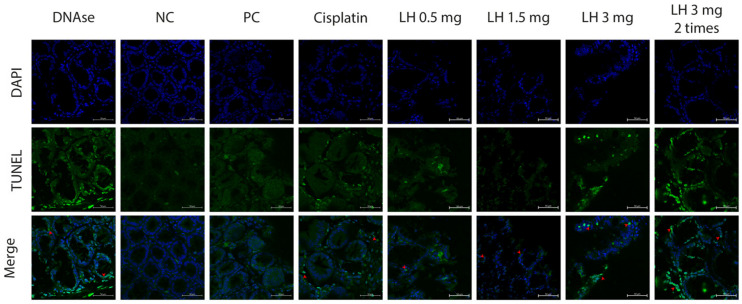
Apoptosis induction by LH (TUNEL assay). Green spots indicate DNA fragmentation and blue spots indicate DAPI staining nuclei. The images were acquired with a confocal microscope with the 40× objective (Zeiss LSM 710 DUO Inverted confocal laser scanning microscope, Oberkochen, Baden-Wurttemberg, Germany). Scale: 50 μm. DNAse corresponds to the positive control of the TUNEL assay. Photos shown are representative of at least three independent experiments. LH concentrations are in mg/Kg.

**Figure 7 cancers-16-00573-f007:**
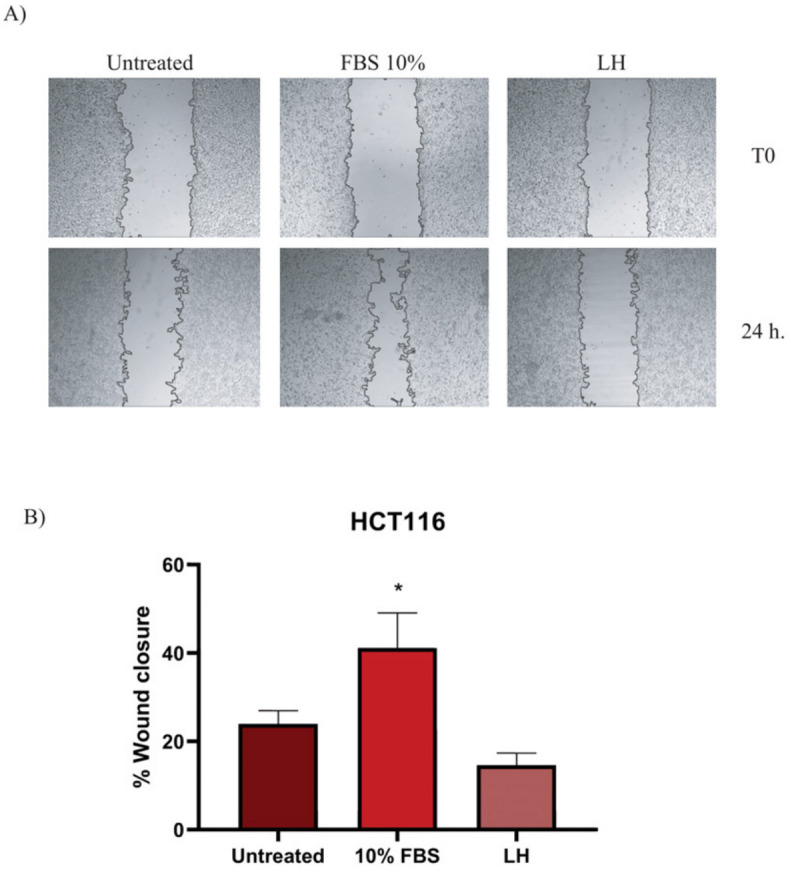
Migration assay in HCT116 cells treated with LH. (**A**) Wound healing assay of cells treated with LH or FBS or not treated at 24 h. (**B**) Percentage of migration at 24 h. * Significant differences against non-treated group. Kruskal–Wallis and post hoc Student Newman–Keuls (*p* < 0.05).

**Figure 8 cancers-16-00573-f008:**
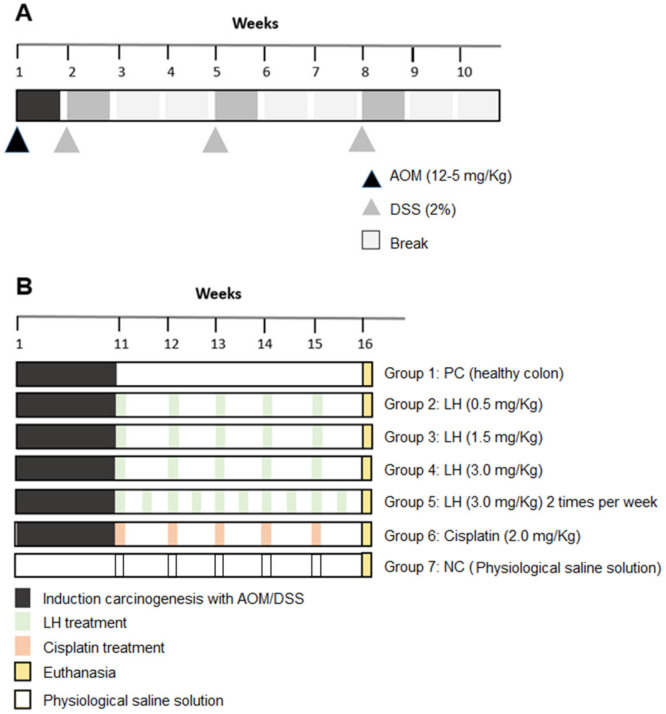
Timeline of the AOM/DSS model. (**A**) Carcinogenesis induction by AOM/DSS. (**B**) Experimental groups treated with LH and cisplatin, NC: negative control, PC: positive control.

**Table 1 cancers-16-00573-t001:** Toxicity analyses in blood in Balb/c mice treated with LH.

Biochemical Analysis	NC	PC	Cisplatin (2.0 mg/Kg)	LH			
				0.5 mg/Kg	1.5 mg/Kg	3.0 mg/Kg	3.0 mg/Kg (Double)
Erythrocytes (×10^12^/L)	9.59 ± 0.34	5.42 ± 0.066 *	5.99 ± 1.65 *	5.29 ± 1.87 *	8.64 ± 1.22	8.86 ± 0.78	8.99 ± 0.18
Leukocytes (×10^9^/L)	7.75 ± 0.77	3.93 ± 0.15 *	3.45 ± 0.77 *	4.2 ± 2.82 *	4 ± 0.7 *	4.26 ± 0.141 *	3.83 ± 0.30 *
Hematocrit (%)	51.1 ± 1.2	27.2 ± 1.6 *	31.2 ± 8.45 *	34 ± 10.3 *	38.66 ±6.57	43.92 ± 3.36	45.6 ± 1.21
Lymphocytes (%)	61.33 ± 6.11	85.33 ± 5.5 *	74.66 ± 3.05 *	86.5 ± 2.12 *	86 ± 8.12 *	82.175 ± 0.82 *	85.66 ± 1.52 *
Neutrophils (%)	34.66 ± 3.5	10.66 ± 7.23 *	23.33 ± 1.15 *	19.33 ± 10.21 *	16 ± 8.5 *	15 ± 1.15 *	13.33 ± 0.57 *
Hemoglobin (g/L)	157 ± 5.2	87.6 ± 4.1 *	98.6 ± 21.7 *	93.6 ± 23 *	120.3 ± 18.4	136 ± 10.7	144 ± 9.5
Glucose (g/L)	1.19 ± 0.09	2.07 ± 0.14 *	0.94 ± 0.09	1.65 ± 0.07 *	1.19 ± 0.09	1.01 ± 0.04	1.08 ± 0.1
Urea (g/L)	0.44 ± 0.02	0.58 ± 0.15 *	0.4 ± 0.06	0.4 ± 0.04	0.41 ± 0.03	0.36 ± 0.02	0.35 ± 0.01
Creatinine (mg/dL)	0.0056 ± 0.00049	0.0064 ± 0.00041	0.0055 ± 0.0007	0.006 ± 0.001	0.005 ± 0.00011	0.0057 ± 0.0012	0.0055 ± 0.0007
Alanine transferase U/I	132.7 ± 0.56	50.86 ± 11 *	89.5 ± 3.5 *	63 ± 2.8 *	110.45 ± 1.76 *	132 ± 3.6	126.33 ± 5.03

NC: negative control (physiological saline solution); PC: positive control (AOM/DSS); LH: laherradurin; Double: two times per week. Data shown are the mean ± SD of three mice per group. * shows statistically significant difference against the negative control. Anova and post hoc Newman–Keuls (*p* ˂ 0.05). Kruskal–Wallis and post hoc Student Newman–Keuls (*p* ˂ 0.05) for results in percentage (hematocrit, lymphocytes, and neutrophils).

## Data Availability

Data are contained within the article.

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
