# Peer review of "Laherradurin Inhibits Tumor Growth in an Azoxymethane/Dextran Sulfate Sodium Colorectal Cancer Model In Vivo"

_cancers, 2024, doi:10.3390/cancers16030573_

Round 1

Reviewer 1 Report

Comments and Suggestions for Authors

This is a nice and very scholarly manuscript. I only have a few minor comments/issues that need to be addressed.

1.       How purity of tested laherradurin was determined? This is crucial for conduction of bioassays from natural products.

2.       Data on table 1 must be presented using the same accuracy. Please, the same number of significant algarisms must be indicated.

3.       you should note that acetogenins are no longer regarded as such promising drug leads in general as they once were, due to their now well-documented association with potential neurological side effects (e.g., Escobar-Khondiker et al., J. Neurosci. 2007, 27, 7827-7837; Le Ven et al., J. Agric. Food Chem. 2014, 62, 8696-7804). Therefore, you will need to include a cautionary note on the potential neurotoxicity of your isolated compounds at the end of the “Results and Discussion” section.

Author Response

Dear Reviewer.

Find attached the corrections to the manuscript

Reviewer 2 Report

Comments and Suggestions for Authors

Dear Authors

This manuscript lacks a great deal of persuasiveness in addition to its numerous deficiencies.

The author's argument for the superiority of LH versus cisplatin is not very compelling.

It is necessary to improve wound healing and TUNEL images. What the image is attempting to convey eludes audience. Authors need to exhibit a clear image.

In Discussion part, the reasoning must be clarified while supplying additional evidence and references.

It is highly advisable to meticulously arrange the entirety of the content in order to enhance the organization and accentuate the clarity of the logical structure.

Comments on the Quality of English Language

Extensive editing of English language required

Author Response

Dear Reviewer

Find attached the corrections to the manuscript

Round 2

Reviewer 2 Report

Comments and Suggestions for Authors

Dear Authors

The manuscript contains some valuable and potential improvement-worthy points; however, I appreciate that you observed the recommendation to revise the inquiries by referencing the repertoire to help readers' comprehension.

However, there is a minor issue with the TUNEL image throughout the entire manuscript. The information on Scale bar is incorrect.

Figure 7, Change the word Laherr to LH.

Comments on the Quality of English Language

Minor editing of English language required

Author Response

Dear Reviewer,

We appreciate the careful reading of our article. Your suggestions are valuable for the improvement of our text. The manuscript has modified accordingly.